# Intraocular RGD-Engineered Exosomes and Active Targeting of Choroidal Neovascularization (CNV)

**DOI:** 10.3390/cells11162573

**Published:** 2022-08-18

**Authors:** Dimitrios Pollalis, Dongin Kim, Gopa Kumar Gopinadhan Nair, Changsun Kang, Arjun V. Nanda, Sun Young Lee

**Affiliations:** 1USC Roski Eye Institute, USC Ginsburg Institute for Biomedical Therapeutics, Department of Ophthalmology, Keck School of Medicine, University of Southern California, Los Angeles, CA 90033, USA; 2Department of Pharmaceutical Sciences, College of Pharmacy, University of Oklahoma Health Science Center, Oklahoma City, OK 73104, USA; 3Department of Ophthalmology, Dean McGee Eye Institute, University of Oklahoma Health Science Center, Oklahoma City, OK 73104, USA; 4College of Medicine, University of Oklahoma Health Science Center, Oklahoma City, OK 73104, USA; 5Department of Physiology, University of Oklahoma Health Science Center, Oklahoma City, OK 73104, USA

**Keywords:** active targeting, choroidal neovascularization, exosome

## Abstract

Purpose: To assess the transretinal penetration of intravitreally injected retinal multicell-derived exosomes and to develop exosome-based active targeting of choroidal neovascularization (CNV) by bioengineering with ASL, which is composed of a membrane Anchor (BODIPY), Spacer (PEG), and targeting Ligands (cyclic RGD peptide). Methods: Retinal multicell-derived exosomes were recovered from a whole mouse retina using differential ultracentrifugation. Their size, number, and morphology were characterized using nanoparticle tracking analysis (NTA) and transmission electron microscopy (TEM). Exosome markers were confirmed using an exosome detection antibody array. Intravitreal injection of fluorescent (PKH-26)-labeled or engineered ASL exosomes (1 × 10^6^ exosomes/μL) were given to the wild-type mouse or laser-induced CNV mouse model. Retinal uptake of exosomes was assessed by in vivo retinal imaging microscopy and histological staining with DAPI, GSA, and anti-integrin *α*_v_ for retinal sections or choroid/RPE flat mounts. Active targeting of CNV was assessed by comparing retinal uptake between areas with and without CNV and by colocalization analysis of ASL exosomes with integrin *α*_v_ within CNV. Staining with anti-F4/80, anti-ICAM-1, and anti-GFAP antibodies on retinal sections were performed to identify intracellular uptake of exosomes and immediate reactive retinal gliosis after exosome treatment. Results: An average of 2.1 × 10^9^ particles/mL with a peak size of 140 nm exosomes were recovered. Rapid retinal penetration of intravitreally injected exosomes was confirmed by retinal imaging microscopy at 3 and 24 h post-injection. Intravitreally delivered PKH-26-labeled exosomes reached inner and outer retinal layers including IPL, INL, OPL, and ONL at 1 and 7 days post-injection. Intravitreally injected ASL exosomes were predominantly delivered to the area of CNV including ONL, RPE, and choroid in laser-induced CNV mouse models with 89.5% of colocalization with integrin *α*_v_. Part of exosomes was also taken intracellularly to vascular endothelial cells and macrophages. After intravitreal injection, neither naive exosomes nor ASL exosomes induced immediate reactive gliosis. Conclusions: Intravitreally delivered retinal multicell-derived exosomes have good retinal penetration, and ASL modification of exosomes actively targets CNV with no immediate reactive gliosis. ASL exosomes have a great potential to serve as an intraocular drug delivery vehicle, allowing an active targeting strategy.

## 1. Introduction

Exosomes are naturally occurring, cell-secreted, and nano-sized (diameter ≈ 30–150 nm) extracellular vesicles (EVs) that steadily carry and transfer different biomolecules throughout the whole body. Exosomes physiologically carry various cargos, including microRNAs (miR), proteins, and lipids for cell-to-cell communication [1,2,3,4]. Based on these characteristics, there are several potential advantages for developing exosome-based intraocular therapy. Because exosomes are naturally produced by all cells, they may be superior to synthetic drug carriers as they provide higher biocompatibility, less toxicity, and better tissue penetration [5,6,7]. Intraocular treatment of the stem cell-derived exosomes that deliver their unique functional cargo has been investigated to reprogram the microenvironment in various ocular pathologies by stem cell-derived paracrine effects [8,9]. Exosomes are also highly engineerable [10,11]. Surface modification of exosomes can confer cell and tissue specificity. Successful exosomal cargo packing has also shown that exosomes can be an excellent natural drug delivery system.

Millions of intravitreal injections of vascular endothelial growth factor (VEGF) neutralizing proteins are performed each year in the United States as the current mainstay treatment for choroidal neovascularization (CNV) in neovascular age-related macular degeneration (NVAMD) [12,13]. However, some patients still experience suboptimal visual outcomes due to insufficient efficacy [14,15]. Several barriers contribute to the suboptimal efficacy of current anti-VEGF treatment. Local delivery of intravitreally injected anti-VEGF agents to the retina primarily depends on diffusion from the vitreous humor to the retina and choroid without active targeting of NV lesions [16]. Biological impedance within vitreoretinal interface-retina-RPE tight junction-choroid is a barrier that limits the local concentration of agents. Frequently, accompanying retinal fibrosis and gliosis secondary to NV are additional barriers to efficient penetration of intravitreally delivered drugs, and currently, there is no treatment for retinal fibrosis [17]. Anti-VEGF monotherapy does not target multiple signaling pathways involved in the pathogenesis of CNV. Clearance of intravitreally delivered anti-VEGF agents through the aqueous and vitreous humor shortens the half-life of these treatments that are resulting in the requirement of frequent intravitreal injection [16].

While exosome-based ocular therapy has been increasingly studied, supporting the potential benefits of their ophthalmological application, exosome-based active targeting strategies in ocular treatment have not been studied [8,9,18]. To employ an active targeting strategy for CNV, we engineered multiretinal cell-derived exosomes modified with ASL, composed of membrane Anchor (BODIPY), Spacer (PEG), and targeting Ligands (cyclic RGD peptide). We previously demonstrated that doxorubicin-loaded ASL exosomes substantially enhanced the anti-cancer efficacy of doxorubicin by active targeting of melanoma where integrin α_v_β_5_ is overexpressed and by reduced systemic clearance of ASL exosomes [19]. RGD is a ligand for a subset of integrin, including integrin α_v_ [19,20,21,22,23,24]. Expression of integrin α_v_ is also increased in ocular tissues from NVAMD and proliferative DR patients [25].

In this study, we evaluated an ASL exosome-mediated active targeting strategy for CNV that has the functionality of delivering multi-molecular targets for posterior eye diseases. We also studied retinal penetration of the intravitreally delivered exosomes and subsequent reactive gliosis after intraocular exosome treatment to assess the potential translation of an exosome-assisted active targeting strategy to treat posterior eye disease.

## 2. Material and Methods

### 2.1. Animals

Wild-type C57BL/6 mice used for the experiments were treated according to the Association for Research in Vision and Ophthalmology guidelines on the care and use of animals in research. All protocols were approved by the Animal Care and Use Committee of the Oklahoma University Health Science Center (OUHSC) (20-055-HL, Approved 30 September 2021) and the University of Southern California (USC) (21415, Approved 26 May 2022).

### 2.2. Exosome Isolation and Characterization

The retinas from 4–6-week-old wild-type C57BL/6J mice were extracted and digested using the Papain Dissociation System (Worthington Biochemical Corporation, Lakewood, NJ, USA) according to the manufacturer’s protocol. Exosomes were recovered using differential ultracentrifugation as previously described [26]. In brief, the supernatant received from the digestion process was mixed with cold HBSS and centrifuged at 25,000× RPM for 60 min at 4 °C in a Beckman Coulter Optima L80-XP ultracentrifuge with rotor SW60 Ti. The supernatant was again subjected to sequential ultra-centrifugation at 41,000× RPM for 60 min at 4 °C for 120 min. The final pellet (exosomes) was resuspended in 100 μL PBS and stored at −80 °C until further use (Figure 1A). The size, number, and morphology of exosomes were characterized by nanoparticle tracking analysis (NTA) (NanoSight, Malvern Instruments Ltd., Malvern, UK) and transmission electron microscopy (TEM), respectively (Figure 1B,D). Exosome markers were analyzed by an exosome detection antibody array (Exo-Check, System Biosciences, Palo Alto, CA, USA) according to the manufacturer’s instructions (Figure 1C). Zeta potential of exosomes was measured using ZetaPALS (Brookhaven Instruments, Holtsville, NY, USA) (Figure 1E).

### 2.3. Fluorescent Labeling of Exosomes

For fluorescent labeling of the exosomes, PKH-26 Red Cell Linker Mini Kit for General Cell Membrane Labeling (Sigma-Aldrich, St. Louis, MO, USA) was used according to the manufacturer’s instructions. In brief, resuspended in 1 mL Diluent C exosome pellets and 1 mL Diluent C mixed with 4 μL PKH-26 were incubated for four minutes. Then, 10% BSA was used to stop the labeling reaction. Labeled exosomes were isolated and purified by sucrose density gradient ultracentrifugation (41,000× RPM for 120 min at 4 °C).

### 2.4. Bioengineering of ASL Exosomes

ASL was synthesized as previously described [19]. In brief, fluorescent lipophilic boron-dipyrromethene (BODIPY), TR-X NHS Ester (7.08 µM) (Thermo Fisher Scientific (Rockford, IL, USA), and TEA (7.08 µM) were completely dissolved in dichloromethane (DCM). Carboxy-PEG_12_-Amine (5.66 µM) (Thermo Fisher Scientific (Rockford, IL, USA) in DCM was added dropwise, and the resulting mixture was stirred at room temperature for 6 h. The reaction mixture was precipitated with cold diethyl ether. Anchor-Spacer-COOH (AS) was obtained as a purple powder. A measure of 6.95 µM of AS, 30.30 µM of 1-Ethyl-3-(3-dimethylaminopropyl) carbodiimide, and 8.34 µM of N-hydroxysuccinimide were dissolved in dimethyl sulfoxide (DMSO), followed by the addition of TEA (8.34 µM). The mixture was stirred for 1 h in the dark. A measure of 10.43 µM of Cyclic Arg-Gly-Asp-D-Phe-Lys (RGD) (Peptide International (Louisville, KY, USA) was completely dissolved in DMSO and added to the mixture. The reaction was kept at room temperature overnight, and the product was purified using the membrane dialysis tubing method (Cutoff MW 1000, Spectrum Laboratories, Inc., Rancho Dominguez, CA, USA). The product was obtained using a sephacryl S-100 HR column (Sigma-Aldrich, St. Louis, MO, USA) and lyophilized. A measure of 7.9 mg of ASL (40 mg/mL) was added to 100 µL of exosomes (1.8 × 10^9^/mL). The ASL exosomes were formulated using a sonication method, and the sample was incubated at 37 °C for 1 h to recover the integrity of the membrane [19].

### 2.5. In Vitro Exosome Uptake

Human retinal microvascular endothelial cells (HRMEC), donated by Dr. Jian-Xing Ma from the University of Oklahoma, were cultivated in an EBM endothelial cell growth medium (Lonza Bioscience, Basel, Switzerland) supplemented with an EGM endothelial cell growth medium SingleQuots kit, including hEGF, hydrocortisone, GA-1000, BBE, ascorbic acid, and FBS (Lonza Bioscience) in a humidified incubator at 37 °C and 5% CO_2_. The cells were treated with either CoCl_2_ (200 μM or 300 μM) or the complete routine medium once they reached confluency of 70–80%. Forty-eight hours later the cells were stained with anti-*α*_v_ for further assessment. Briefly, the cells were fixed in 4% PFA and blocked in 10% horse serum. Then, they were stained with anti-*α*_v_ integrin (SC9969, Santa Cruz, CA, USA) primary antibodies. The secondary antibody used was anti-mouse Alexa Fluor 594 (A11005, Invitrogen, Waltham, MA, USA) and 4′6′-diamino-2-phenylindole (DAPI, H-1500, Vector Laboratories, Newark, CA, USA) was used for nuclei visualization. Using HRMECs cultured as described above, ASL exosome cellular uptake was also evaluated. After confluence reached 70–80%, either CoCl_2_ or the regular routine medium was applied for 24 h. ASL exosomes were then added, and cells were incubated for a further 24 h. After a total 48-h incubation they were fixed and stained with anti-*α*_v_ integrin antibodies.

### 2.6. Laser-Induced Choroidal Neovascularization Mouse Model

Laser photocoagulation was performed on wild-type C57BL/6J 4–6-week-old mice [27]. Briefly, pupils were dilated using 2.5% phenylephrine hydrochloride and 1% cyclopentolate hydrochloride. Mice were anesthetized with ketamine (100 mg/mL) and xylazine (100 mg/mL) via intraperitoneal (IP) injection. Laser photocoagulation was performed using the Micron IV retinal imaging system (Phoenix Research Lab, Pleasanton CA, USA) with the Meridian Merilas 532 green laser (50 μm, 70 ms, 240 mW). Three lesions were induced located approximately one disk diameter from the optic nerve with respect to the large vessels. Laser-induced disruption of Bruch’s membrane was confirmed by the appearance of a bubble at the site of photocoagulation. Induction of CNV was confirmed by fundus photography, optical coherence tomography (OCT), fluorescent angiography, and H&E stain of the retinal section seven days after laser photocoagulation.

### 2.7. Intravitreal Injection of Exosomes

C57BL/6J 4–6-week-old mice (*n* = 9) received intravitreal injection of PKH-26 labeled exosomes or ASL exosomes. Laser-induced CNV mouse models received intravitreal injection exosome treatment three days after laser photocoagulation. For intravitreal injection, mice were anesthetized with IP ketamine (100 mg/mL)/xylazine (100 mg/mL) and pupils were dilated with 2.5% phenylephrine/1% cyclopentolate. A small scleral incision was made posterior to the limbus using a 31-gauge (G) insulin needle. Then, a 33G blunt needle attached to a 10 μL NanoFil syringe (World Precision Instruments, Sarasota, FL, USA) was used to deliver 1 μL (1 × 10^6^ exosome particles/1 μL) of solution in the vitreous cavity through the same incision.

### 2.8. Intravenous Injection of Exosomes

C57BL/6J 4–6-week-old mice (*n* = 2) received an intravenous injection of PKH-26-labeled exosomes through the tail vein (5 × 10^7^ exosome particles/50 μL). Mice were euthanized three hours after injection, and their eyes were harvested to study the retinal uptake of systemically administered exosomes.

### 2.9. In Vivo Imaging Analysis

Fundus photography, in vivo retinal imaging microscopy, and fluorescein angiography (FA) were performed using the Micron IV retinal imaging system (Phoenix Research Lab, Pleasonton, CA, USA). Mice were anesthetized with IP ketamine (100 mg/mL)/xylazine (100 mg/mL) and pupils were dilated with 2.5% phenylephrine/1% cyclopentolate. For FA, mice received an intraperitoneal injection of 10% of fluorescein sodium (AK-Fluor, Akorn, Lake Forest, IL, USA). Optical coherence tomography (OCT) was performed using OCT (Bioptigen, Durham, NC, USA).

### 2.10. Histological Analysis

The mice were euthanized seven days after the intravitreal injection of the exosomes, and the eyes were harvested for either flat-mount histology or cryopreserved for retinal sections (8 μm). For flat mount immunostaining, the cornea, lens, vitreous, and retina were removed, and the RPE/choroid complex was fixed in 4% PFA for 3 h. Then, the tissue was blocked in 10% horse serum, stained with Rhodamine-conjugated GSA (RL-1102, Vector Laboratories) overnight, and flat-mounted. For cryosection immunohistochemical staining, eyes were fixed in 4% PFA, embedded in an optimal cutting temperature medium (Sakura, Japan), frozen with liquid nitrogen, and stored at −20 °C. Leica CM3050S was used to obtain 8 μm thick frozen sections. The sections were stained with anti-*α*_v_ integrin (SC9969, Santa Cruz, Dallas, TX, USA), anti-F4/80 (SC 377009, Santa Cruz, Dallas, TX, USA), and anti-ICAM-1 (AF796, R&D Systems, Minneapolis, MN, USA), with the primary antibodies and GSA-lectin rhodamine labeled (RL-1102, Vector Laboratories Burlingame, CA, USA). The secondary antibodies used were anti-mouse Alexa Fluor 594 (A11005, Invitrogen), anti-rabbit Alexa Fluor 594 (A11037, Invitrogen, Waltham, MA, USA), anti-rabbit Alexa Fluor 488 (A11034, Invitrogen, Waltham, MA, USA), anti-goat Alexa Fluor 555 (A21432, Invitrogen, Waltham, MA, USA), and 4′6′-diamino-2-phenylindole (DAPI, H-1500, Vector Laboratories, Burlingame, CA, USA) for nuclei staining. Stained RPE/choroid flat mounts and frozen sections were examined using Olympus FluoView FV1200 confocal microscope.

### 2.11. Image Analysis

FIJI was used for image analysis. For integrin *α*_v_ expression, z-stacks of the RPE/choroid flat mounts were used. The acquired z-stacks were converted to binary stacks, and the threshold areas were used for volumetric analysis using the formula: $V=h\times \sum_{i=1}^{N}A_{i}$, *V*: total volume, *h*: interval between z-stack sections, *A*: threshold area. The JACoP plugin was applied for colocalization analysis, and Pearson’s and Manders’ coefficients were reported and evaluated.

### 2.12. Statistical Analysis

Unless otherwise described, all values were reported as mean ± standard deviation (SD). GraphPad Prism was used for the statistical analysis and graph plotting. Statistical differences were measured with Student’s *t*-test to compare the two groups. JMP Pro 14 was used for graph plotting. *p* values < 0.005 were considered to be significant.

## 3. Results

### 3.1. Exosome Recovery and Characterization

Exosomes were recovered from the whole retina of the 4–6-week-old wild-type (C57BL/6) mice with 2.1 × 10^9^ particles/mL with a peak size of 140 nm. Exosome markers were positive, including FLOT1, ICAM, ALIX, CD81, CD63, ANXA5, and TSG101, and the morphology of exosomes was confirmed by TEM (Figure 1).

### 3.2. Retinal Uptake of Intravitreally Delivered Exosomes and BRB crossing of Systemically Delivered Exosomes

In vivo retinal imaging microscopy confirmed that PKH26-labeled fluorescent exosomes started to be taken from the vitreous humor to the retina at 3 h post intravitreal injection, and mostly, they were brought into the retina in a scattered fashion 24 h post intravitreal injection (Figure 2A). Confocal microscopy of flat-mounted retinas (B) and retinal sections (C) obtained one and seven days after the intravitreal injection of exosomes showed that exosomes were taken into both the inner and outer retina in a scattered fashion (Figure 2B,C). Systemically administered PKH-26 labeled exosomes via the tail vein were observed within the retina 3 h after intravenous injections of exosomes, suggesting that exosomes bypassed the blood-retinal barrier (BRB) (Figure 2D).

### 3.3. Engineered Anchor, Spacer, and RGD Ligand Modified ASL Exosomes

Multiple retinal cell-derived exosomes were modified with membrane Anchor (BODIPY), Spacer (PEG), and targeting Ligands, and cyclic Arg-Gly-Asp (RGD) peptide (Figure 3A). A peak size of 125 nm of ASL exosomes was generated and confirmed by NTA (Figure 3B). The morphology of ASL exosomes was also confirmed by TEM (Figure 3C) and the Zeta potential of ASL exosomes was measured (Figure 3D). Fluorescent (green) BODIPY allowed for the study of the biodistribution of intravitreally delivered ASL exosomes.

### 3.4. Increased Integrin α_v_ Expression in Choroidal Neovascularization

Reliable CNV induction in laser-induced CNV mouse models was confirmed by fundus photography, optical coherence tomography (OCT), fluorescent angiography, and H&E staining of the retinal section seven days after laser photocoagulation (Figure 4A). Expression of integrin α_v_ in the retina of the laser-induced CNV mouse model was increased by over 100 times compared with expression of integrin α_v_ in the non-lasered retina (Figure 4B).

### 3.5. Active Uptake of ASL Exosomes to CNV Sites

In the laser-induced CNV mouse retina, intravitreally delivered ASL exosomes were predominantly found in CNV lesions and rarely seen in non-CNV lesions. This was confirmed in both choroid/RPE flat-mount and retinal sections. In retinal sections, ASL exosomes were observed at the site of CNV in outer retinal layers including ONL, subretinal space, RPE, and choroid. Immunohistochemistry (IHC) of the retinal sections showed 89.5% of colocalization between fluorescent ASL exosomes and integrin α_v_ within CNV lesions (Figure 5A–D). HRECs exposed to hypoxic stress induced by CoCl_2_ in vitro increased the expression of integrin α_v_ (Figure 6A) and the intracellular uptake of ASL exosomes (Figure 6B).

Some of the local delivery of exosomes to CNV sites in vivo was done intracellularly. IHC with anti-ICAM1 (vascular endothelial cell marker) and anti-F4/80 (macrophage marker) antibodies to determine the identification of these cells showed that ASL exosomes were taken up by both vascular endothelial cells (D) and macrophages (C) (Figure 6C,D).

### 3.6. No Reactive Retinal Gliosis after Intravitreal Injection of ASL Exosomes

GFAP expression was not increased within the retina after an intravitreal injection of either naïve exosomes (1 day) or ALS-exosomes (7 days), suggesting that intravitreal injection of naïve or ASL-modified exosomes did not induce an immediate reactive retinal gliosis (Figure 7).

## 4. Discussion

In this study, we describe ASL composed of the Anchor, Spacer, and Arg-Gly-Asp acid (RGD) Ligand modification as an uncomplicated modification of exosomes to promote active targeting of CNV. We developed an exosome-based novel targeted strategy for the treatment of CNV that has many advantages over synthetic nanoparticles, including (1) full-thickness retinal penetration of the retinal multicell-derived exosomes delivered by intravitreal injection as well as bypassing the blood-retinal barrier (BRB), (2) highly efficient active targeting of CNV by RGD-modified exosomes, (3) multifaceted exosome uptake to extracellular and intracellular spaces including retinal vascular endothelial cells and macrophages, and (4) no immediate reactive gliosis from the intravitreal injection of retinal multicell line-derived exosomes or ASL-modified exosomes.

We showed that exosomes administered by intravitreal injection reached the retina, were taken to the inner and outer retina, as well as near the RPE, supporting that intravitreal injection is a good route for retinal penetration in intraocular exosome therapy. Our data support the utilization of exosomes as an intraocular cargo delivery carrier because penetrating the ocular barriers is one of the significant obstacles to the application of thoroughly well-studied synthetic nanocarriers such as polymeric nanoparticles, and liposomes for the treatment of posterior eye diseases. For example, subretinal injection of nanocarriers was often necessary to achieve cargo deliveries to photoreceptors, RPE, or choroids for efficient penetration. However, subretinal injections requiring an intraoperative procedure are more demanding than office-based intravitreal injections. We also confirmed that systemically administered exosomes could cross the blood-retinal barrier (BRB). It has been shown that exosomes can cross natural tissue barriers such as the blood-brain barrier (BBB), and there has been speculation that exosomes will pass the BRB [28,29]. Initially, the size of the nanoparticles was recognized as one of the key factors in tissue penetration, and especially in the bypassing of natural tissue barriers. With an improved understanding of the mechanism of tissue uptake in exosomes, there has been increasing evidence that exosome origin, cargo load, and surface protein may have a more significant influence on tissue uptake of exosomes [30,31]. For example, the molecular composition of exosomes, such as cell surface protein or exosomal cargo, reflects their cell of origin, and cellular or tissue tropism has been shown in exosome uptake [30,31]. While there are only a few reports with a comprehensive study of retinal uptake of intravitreally delivered exosomes; Mathew et al. previously showed that intravitreally delivered bone marrow-derived mesenchymal stem cell-derived exosomes were rapidly cleared from the vitreous and taken to the retina, however exosomes were found mostly no deeper than INL and they were predominantly taken from RGC and microglial cells [32]. Different retinal penetration between previous studies and our study may be due to the different cellular origination of the exosomes. In our study, we tested age matched (4–6-week-old) multiretinal cell-derived exosomes to optimize the potential benefit of favorable retinal penetration of exosomes. To translate intraocular exosome therapy, further studies are needed to determine the source of exosomes.

Although exosomes are naturally occurring cell-secreted nano-sized extracellular vesicles, exosomes are also highly engineerable like synthetic nanocarriers. To establish exosome-based active targeting, we have developed an ASL system composed of an Anchor (BODIPY), Spacer, and Arg-Gly-Asp acid (RGD) Ligand modification. In our previous study, we demonstrated that a fluorescent lipophilic boron-dipyrromethene (BODIPY), used as an exosome anchor, benefited the biodistribution and pharmacokinetic study of ASL exosomes and provided a conjugate domain for a spacer [19]. A PEG spacer conjugated to BODIPY anchors prevented exosome fusions and aggregation at higher concentrations and minimized systemic clearance [19]. RGD is one of the major integrin-binding ligands (Figure 3C) [20,21,22,23,24]. Integrins are essential in VEGF signaling in ocular NV. Although a few previous studies reported that RGD conjugation could actively target CNV, applying synthetic nanoparticles to conjugate with RGD or systemic administration of these particles diminished their clinical utility [33,34,35,36]. Here, we demonstrated that multiretinal cell-derived exosomes modified with an ASL system actively targeted CNV in a laser-induced CNV mouse model by promoting their uptake to overexpressed integrin α_v_ at the site of CNV. In contrast, naïve exosomes penetrated the inner and outer retina in a scattered fashion. Applying an active targeting strategy for CNV can potentially improve the therapeutic efficacy (1) by increasing the local concentration of therapeutic cargo and (2) by enhancing the penetration of anatomical barriers such as retinal fibrosis or gliosis at the site of CNV through the direct intravitreal injection of therapeutic agents, which may overcome the limitation of current intravitreal injection of anti-VEGF agents. Previously increased expression of integrins α_v_ has been shown in human ocular tissues with CNV and proliferative diabetic retinopathy (PDR) patients, further supporting that an RGD-mediated active targeting strategy for CNV may benefit the treatment of a broad spectrum of retinal neovascularization such as diabetic retinopathy and retinal vein occlusion [25].

In our study, we further confirmed that some of the intravitreally delivered ASL exosomes were taken up intracellularly at the site of CNV lesions. These cells included vascular endothelial cells and macrophages within CNV lesions. Our data support that ASL exosomes have great potential to deliver therapeutic cargo both extracellularly (e.g., cell membrane) and intracellularly. This result suggests that ASL exosomes can be utilized for multimolecular and extracellular or intracellular target treatments. For example, ASL exosomes can pack with multiple cargos such as anti-VEGF and microRNAs (miR), which can potentially increase the therapeutic efficacy of CNV treatment compared to anti-VEGF monotherapy.

Despite the perceived low immunogenicity of exosomes due to their physiological origin, the immunogenicity of any recombinant compound needs to be carefully tested prior to its application to intraocular treatment. In our study, we confirmed that neither naïve exosomes nor ASL exosomes delivered by intravitreal injection induced immediate reactive retinal gliosis. While evaluating the immune response, intraocular exosome treatments from various engineering processes will be needed before translating intraocular exosome treatment in humans; our data is encouraging for the advancement of exosome-based active targeting strategies for the treatment of CNV and other retinal neovascular diseases.

The present study had limitations. ASL exosome-mediated active targeting of CNV was only tested in a fully developed CNV in a laser-induced CNV mouse model. The efficacy of the active targeting of ocular NV needs to be tested in the early stages of CNV as well as other types of ocular NV such as diabetic retinopathy. For the translational application of intraocular exosome therapy coupled with an active targeting strategy, further studies in the cellular tropism of exosome uptake in retinal tissues are needed to determine the source of the exosome for human application. Optimizing intended cargo packing into the exosomes and pharmacokinetics are also needed for future translation and the application of intraocular exosome therapy.

## Figures and Tables

**Figure 1 cells-11-02573-f001:**
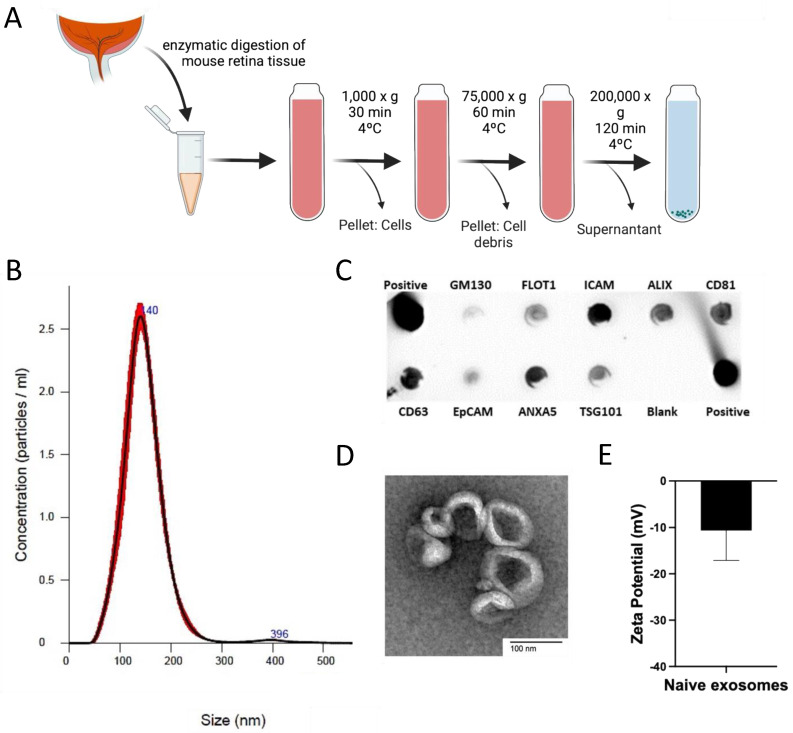
**Recovery and characterization of retinal multicell-derived exosomes.** Exosomes were recovered from the whole mouse retina by differential ultracentrifugation (**A**). The average peak size of exosome particles (140 nm) and an average number of particles of exosomes (2.1 × 10^9^ particles/mL) was determined by nanoparticle tracking analysis (NAT) (**B**). Exosome markers including FLOT1, ICAM, ALIX, CD81, CD63, ANXA5, and TSG101 were confirmed by exosome detection antibody array (**C**). The morphology of exosomes was confirmed by transmission electron microscopy (TEM) (**D**) and Zeta potential of exosomes was measured (**E**).

**Figure 2 cells-11-02573-f002:**
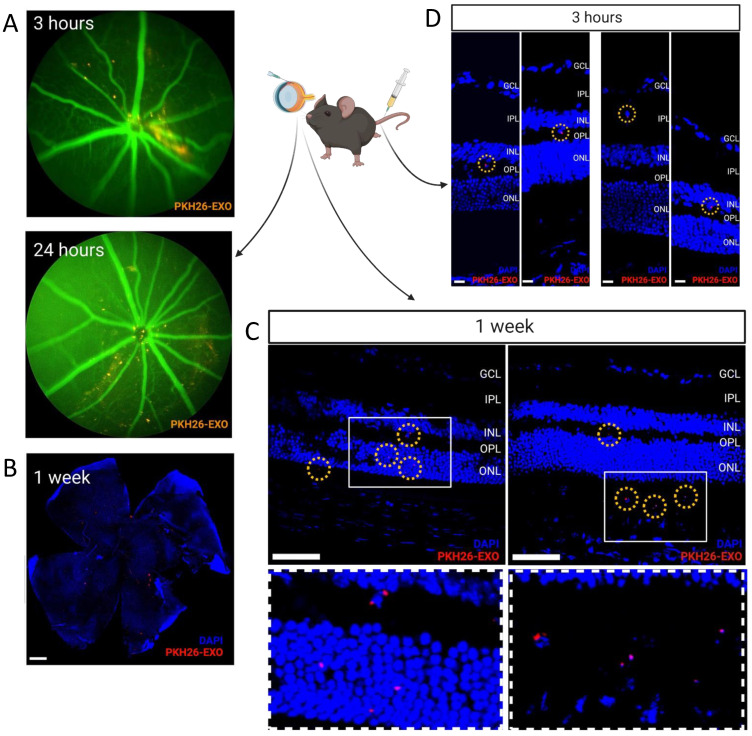
**Retinal uptake of intravitreally delivered exosomes and BRB crossing of systemically delivered exosomes.** In vivo retinal imaging microscopy performed at 3 and 24 h and retina flat-mounts obtained 1 day and 1 week after intravitreal injection of PKH-26-labeled exosomes (orange-red) show immediate retinal penetration and wide distribution of intravitreally delivered exosomes (**A**,**B**). Retinal sections (8 μm) obtained 24 h after intravitreal injection of PKH-26-labeled exosomes (ret dots with yellow circles) shows that exosomes were taken up by both the inner and outer retina (Scale bar, 500 μm) (**C**). Retinal sections (8 μm) obtained 3 h after systemic injection of PKH-26-labeled exosomes (ret dots with yellow circles) via tail vein injection shows that exosomes were taken up by the retina, suggesting that exosomes can cross the blood-retina barrier (BRB). (Scale bar, 10 μm) (**D**).

**Figure 3 cells-11-02573-f003:**
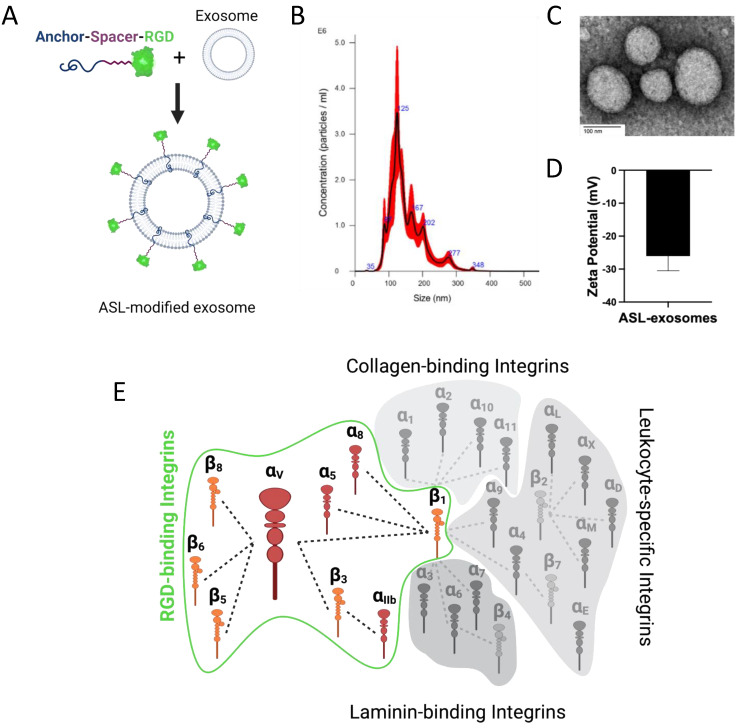
**Engineering of Anchor, Spacer, and RGD Ligand (ASL)-modified exosomes, ASL exosomes.** Retinal multicell-derived exosomes were engineered with ASL modification composed of a membrane Anchor (BODIPY), Spacer (PEG), and targeting Ligand, and cyclic Arg-Gly-Asp (RGD) peptide (**A**). The size and morphology of ASL exosomes were confirmed by NTA (mean size: 152 nm) (**B**). RGD is one of the major binding ligands for integrin families including α_5_β_1_, α_8_β_1_, α_V_β_1_, α_V_β_3_, α_V_β_5_, α_V_β_6_, α_V_β_8_, and α_IIb_β_3_ (**C**). Zeta potential of exosomes was measured (**D**). Major integrin binding ligands (**E**).

**Figure 4 cells-11-02573-f004:**
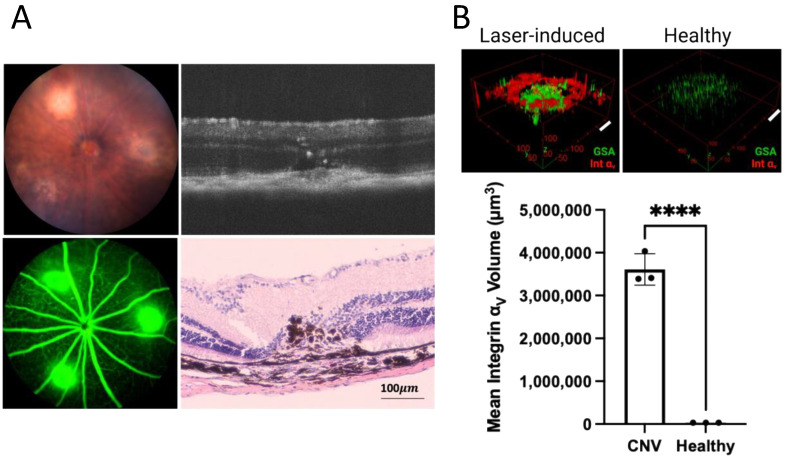
**Increased integrin**α**_v_ expression in choroidal neovascularization in a laser-induced mouse model.** CNV induction in a laser-induced mouse model was confirmed by color fundus photography, fluorescein angiography, OCT, and H&E histology (**A**). Volumetric analysis of integrin α_v_ expression within CNV lesions (*n* = 3) shows an over 100 times increase in integrin α_v_ expression within CNV lesions compared with the control retina, scale bar, 100 μm (bar represents mean ± SD, **** *p* < 0.0001) (**B**).

**Figure 5 cells-11-02573-f005:**
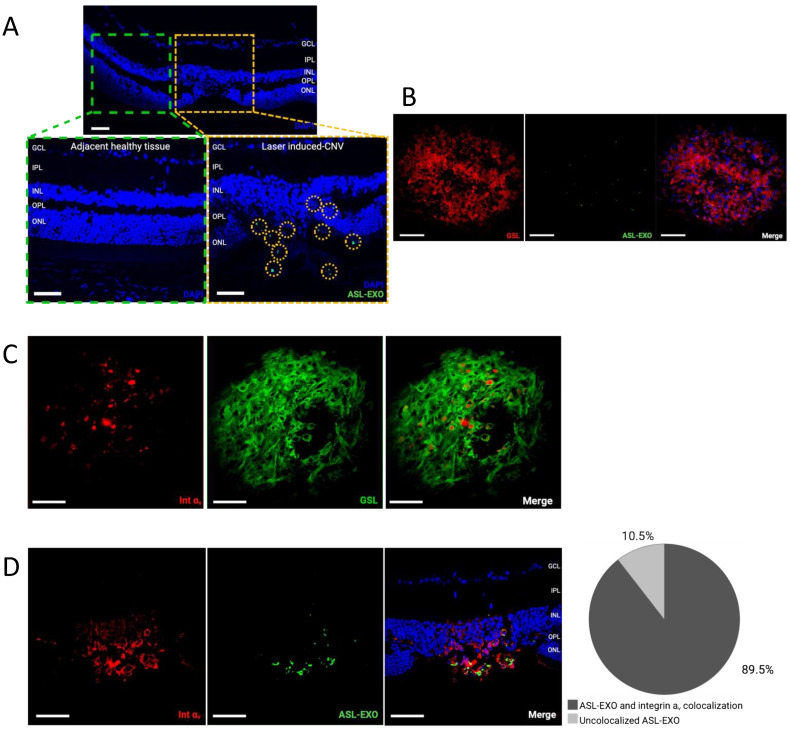
**Intravitreally injected ASL exosomes were primarily delivered to the site of CNV.** Intravitreal injection of ASL exosomes (1 μL) was given 3 days after laser treatment in a mouse model of CNV and the eyes were harvested 7 days after injection to obtain retinal sections (8 μm) and RPE/choroid flat-mounts. Representative eye sections showing ASL exosomes (green) exclusively delivered to the area of CNV (**A**). Representative RPE/choroid flat-mount of a CNV area showing ASL exosomes (green) delivered to the newly formed vessels (rhodamine-conjugated GSL, red) (**B**) and increased integrin α_v_ expression (red) within CNV area (rhodamine-conjugated GSL, green) (**C**). The ASL exosomes (green) are colocalized with integrin a_v_ (red) with 89.5 % of overlapping (*n* = 3) (**D**), scale bar, 100 μm..

**Figure 6 cells-11-02573-f006:**
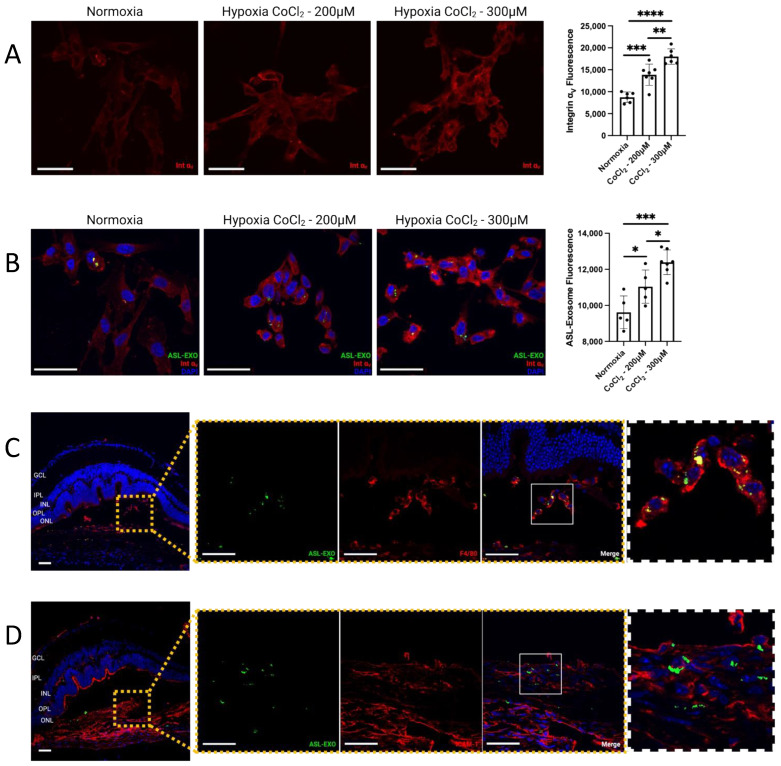
**Intracellular uptake of ASL exosomes.** Human retinal endothelial cell (HRECs) exposed to hypoxic stress induced by either 200 μM or 300 μM CoCl_2_ increased expression of integrin α_v_ (**A**) and increased intracellular uptake of ASL exosomes in vitro (**B**) (bar represents mean ± SD, * *p* < 0.05; ** *p* < 0.01; *** *p* < 0.001; **** *p* < 0.0001). Intravitreal injection of ASL exosomes (1 μL) was given 3 days after laser treatment in a mouse model of CNV and the eyes were harvested 7 days after injection to obtain retinal sections (8 μm) and RPE/choroid flat-mounts. Representative eye sections showing ASL exosomes (green) internalized by macrophages (F4/80, red) (**C**) and endothelial cells (ICAM-1, red) (**D**), respectively, at the area of CNV in vivo. Scale bar, 50 μm.

**Figure 7 cells-11-02573-f007:**
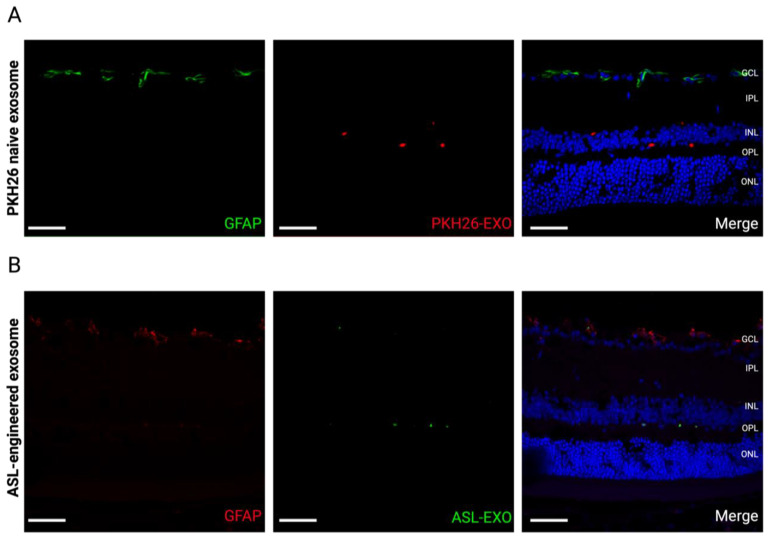
**No immediate reactive retinal gliosis by intraocular exosome treatment.** Retinal sections show no immediate reactive retinal gliosis stained with GFAP antibody staining 1 day after intravitreal injection of PKH-26-labeled exosomes (**A**) and 7 days after the intravitreal injection of ASL exosomes (**B**) (Scale bar, 50 μm). Note that GFAP is only expressed within ganglion cells in a physiologic state.

## Data Availability

This study did not report any data.

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
