# Peer review of "Intraocular RGD-Engineered Exosomes and Active Targeting of Choroidal Neovascularization (CNV)"

_cells, 2022, doi:10.3390/cells11162573_

Round 1

Reviewer 1 Report

In this manuscript, exosomes from multiple retinal cell sources were collected from mouse retinas. An exosome modified by membrane anchoring (BODIPY) - spacer (PEG) - targeting ligand (cyclic RGD peptide) (ASL) was designed, and an exosome based nano carrier platform for actively targeting choroidal neovascularization (CNV) was developed. The laser-induced CNV mouse model was successfully constructed. After intravitreal injection of fluorescent (pkh-26) labeled ASL exosomes, its active targeting in the mouse model was evaluated according to the uptake in the diseased and non diseased areas, and its good retinal permeability was proved. GFAP staining proved that the injection of exosomes or engineered exosomes in vivo would not produce immediate reactive gliosis. It is interesting yet there are some issues should be considered before its publication.

(1) In the aspect of exosome characterization, there is a lack of data support of conventional zeta potential and the successful construction of engineered exosomes.

(2) BODIPY can be used as a biodistribution and pharmacokinetic marker, so ASL modification not only enhances the stability of active targeting of exosomes, but also introduces a built-in biological imaging mode. Why not apply it to this for CO localization analysis,

(3) Lack of support for cell experiments, such as low expression α v β 3 human umbilical vein endothelial cells and high expression α v β 3 human umbilical vein endothelial cells as the target cells, and the targeting of exosomes was verified by in vitro vascular endothelial cell tube formation test. ASL exosome mediated active targeting of CNV was also tested only in laser-induced CNV mouse models in fully developed CNV.

(4) The drug loading efficiency was not further evaluated, thus lacking of persuasion that it can become a drug carrier for ocular fundus neovascular diseases.

(5) In vivo biocompatibility data support is also less, (exosomes play a protective role in various physiological processes, including vascular repair and regeneration. Experiments only prove that the engineered exosomes are highly ingested and will not cause gliosis, but do not indicate that the engineered exosomes have no angiogenesis promoting effect by themselves,) and do not indicate whether the engineered exosomes will produce cytotoxicity.

Author Response

We appreciate the constructive comments. We attached our response to reviewer 1. 

Reviewer 2 Report

In this study, Pollalis and colleagues investigate the distribution of f intravitreally injected retinal-derived exosomes. They define the isolation procedure of retinal exosomes and proceed to characterize them. The show that these exosomes can be found in the retina after both intravitreal and intravenous delivery. Using a model of CNV, they also show that these exosomes accumulate preferentially in vascular lesions, where they colocalize specifically with integrin ?v. Finally, they show that delivery of retinal exosomes does not appear to cause retinal gliosis.

Overall, this is an interesting, if descriptive, study which highlights the potential of retinal-derived exosomes as carriers for therapeutic molecules to the retina. I would however suggest some revisions to improve the current study:

-          The results section is not very detailed, and some experimental details are missing (age of mice, timing of experiments, etc.). There is also limited rationale given for some experiments in this section and often results are described, but without context which makes this section a bit disjointed. As such, I believe that the results section should be reworked to improve clarity.

-          I would modify the title provided to remove therapy, as while the study shows exosome accumulation at the site of CNV lesions, no therapeutic effects of these exosomes are evaluated.

-          How old were the mice from which retinas used for exosome isolation were used? Does age affect exosome recovery? One could imagine that exosome markers may differ between ageing mice and mouse pups. This should be clarified, and the age of mice used justified.

-          In Fig.2B, PKH-labeled exosomes seem to localize to a restricted part of the retina after 1 day rather than being scattered throughout the retina. Why is that?

-          In the images shown in Fig.2C, it is actually difficult to see the PKH fluorescence in the images shown. It would be easier to see if micrographs without DAPI were also provided.

-          In Fig. 4B, what cell types are being labeled by integrin ?v? The staining does not seem to colocalize with GSA, which would imply that its expression is outside of the vasculature?

-          From the micrographs shown in Fig.5, I am not overly convinced that exosomes are really uptaken by endothelial cells. While there appears to be colocalization of ASL-exosomes and F4/80, the fluorescence of ASL-exosomes exosomes does not match with that of the ICAM-1 staining. Furthermore, integrin ?v staining does not quite match GSA staining in Fig.4 and Fig.5, suggesting that this integrin may be expressed in non-endothelial cells in the CNV lesions. Double immunostaining of F4/80 and integrin ?v should be provided, as it seems more likely this integrin subunit is expressed in macrophages, thus explaining why the uptake of ASL-exosomes is more prominent in these cells.

-          In the experiments displayed in figure-5, how long after exosome delivery were retinas/choroids tissues evaluated?

Author Response

We appreciate the constructive comments and attached our response to reviewer #2. 

Round 2

Reviewer 1 Report

The revision is better than the previous manuscript draft. 

Reviewer 2 Report

The manuscript has significantly been improved, and concerns were addressed.